# Developing A Model of Mobility Capital for An Ageing Population

**DOI:** 10.3390/ijerph16183327

**Published:** 2019-09-10

**Authors:** Charles Musselwhite, Theresa Scott

**Affiliations:** 1Centre for Innovative Ageing, Swansea University, Swansea SA2 8PP, UK; 2School of Psychology, University of Queensland, St Lucia, QLD 4072, Australia

**Keywords:** ageing, older people, later life, mobilities, social capital, cultural capital, transport, giving up driving, driving cessation

## Abstract

Driving a car meets older people’s needs, providing utility (getting from A to B), psychosocial (providing identity and roles and feelings of independence and normality) and aesthetic (mobility for its own sake) mobilities. Giving up driving is related to poorer health and wellbeing. This paper addresses how older people cope when they give up driving, using Bourdieu’s theory of capital as a way of categorising different barriers and enablers to managing without a car in a hypermobile society. Older people are most likely to mention barriers and enablers to mobility relating to infrastructure capital (technology, services, roads, pavements, finance and economics), followed by social capital (friends, family, neighbourhood and community). Cultural capital (norms, expectations, rules, laws) and individual capital (skills, abilities, resilience, adaptation and desire and willingness to change) are less important but still significantly contribute to older people’s mobility. Implications for policy and practice suggest that provision for older people beyond the car must explore capital across all four of the domains.

## 1. Introduction

### 1.1. Context

Countries across the globe are experiencing rapid ageing due to a combination of people living longer because of better health and social care, and lower birth rates, resulting in both higher numbers of older people and a higher percentage of older people within society. Across the globe, there are almost 900 million people over 60, making up 11.7 percent of the population [1]. By 2050, the world’s population aged 60 years and older is expected to total 2 billion, and will make up 22% of the population [1]. The global population is ageing, however western societies have aged more rapidly and, for example, 25 percent of the UK’s population is likely to be over 60 by around 2030 [2].

Older people are also more likely than ever before to be active and mobile. In Western Societies this is because of what is termed a hypermobile society, where people’s activities take place over long geographical distances, accessed through a motorised private vehicle such as a car. Hospitals, shops and services are placed at locations most accessible by the car. People are more likely to move around for work and live away from extended family. In later life, people have connections to others, to services and shops, across a wide geographical location. Older people have a strong preference for and attachment to their car. The car is an easy way of connecting to these spaces. However, ordinary ageing processes, changing cognition, health conditions, eyesight and physiology mean driving becomes more difficult for older people [3,4]. Consequently, often they have to give up driving, though it is noticeably hard to do so because of a lack of perceived alternatives and the perceived control and convenience the car provides, as well as affective and emotional connections to the vehicle. This paper addresses how older people cope when they give up driving in a hypermobile society. It looks as how different resources within an individual’s community or neighbourhood can be a barrier or can help the mobility of an individual who does not drive. It takes the concept of Bourdieu’s theory of capital as a way of categorising these different barriers and enablers to coping with life without a car.

### 1.2. Literature

Musselwhite and Haddad [5,6] place older people’s mobility needs in a three-level hierarchical model. The three levels are equally important, but they are based in the order of how explicitly the need was discussed by older people. The order also tends to reflect the priority of interventions aimed at improving mobility for older people, with concentration at the base and less support for the middle and higher-level needs [7]. At the base of the hierarchy is utility (representing literal mobility, journeys to get from A to B as conveniently and reliably as possible), which is where older people place most importance and where services and provisions are placed for those who have given up driving. At the next level up, with less attention from older people and support services is the affective level of mobility, the need for independence, freedom and to have status and control over life that mobility affords. At the top level, which is no less important but often neglected, is the need for travel for its own sake, to get out and about and see the world, to experience travel itself. Musselwhite suggests the car meets all three levels of need better than other modes of transport, and hence giving up driving is problematic as older people do not get all of their needs met [8]. Not surprisingly therefore, giving up driving is often described as a major life event, related to reduced quality of life and poorer health and wellbeing [9,10,11,12,13,14,15,16,17]. However, some people give up driving “successfully”, that is without major reduction in quality of life and manage to maintain their health and wellbeing. This paper will examine what strategies people employ to mitigate the negative effect of giving up driving.

One of the major reasons that people are tied to driving in later life is lack of alternatives. Musselwhite suggests a model of age-friendly transport that covers, in concentric rings around the individual, walking and cycling followed by public and community transport, and then all surrounded by age-friendly strategy and policy [18]. To successfully negotiate mobility in later life post-car therefore, older people must have provision in all these levels of support.

To replace journeys by walking or cycling, although they have significant health benefits [19], is problematic. At a macro level, as services and shops get placed at the edge of town or city centres on major road links, journeys that could have been taken by foot or by bicycle in previous years are no longer possible. At a microscopic level, the poor infrastructure can prohibit older people from walking, including cracked and poorly maintained pavements, lack of pavements altogether and poor crossing facilities, including formalised crossings that do not allow enough time for older people to cross [20,21]. Public and community transport can offer some alternative to car driving. But provision of bus services is often sporadic, especially in rural areas, often geared around commuting or transporting people for work, something less of importance to older people. In addition, there are public concerns for safety, with bus drivers driving off before the older person has sat down, or passengers not having level access when boarding and alighting [22,23]. Provision of community transport, specialist transport services for those unable to access traditional bus services, are again sporadic in availability and can be difficult to use, with cumbersome booking systems and lengthy journeys to pick up and set down people at their front doors [23]. Transport policies rarely have an ageing focus, with models focused around commuting and work-based trips, which are less likely to be of importance to older people who, at present, are less likely to be in full-time employment [24].

Murray and Musselwhite discussed the importance of informal support provided to an older person as they go through the process of giving up driving [25]. Support was often provided in very practical terms (for example, provision of a lift or obtaining goods for the person), but the receiver would often also need psychosocial and social support (for example emotional support). Support was more likely to occur in areas with no bus service. However, it is common for the older people seeking support to feel a burden on others. Consequently they reduced that feeling by rationing their need for help, through reducing times they go out and trip chaining, as well as often providing reciprocation through gifts of money or other support.

This paper attempts to categorise older people’s ability to overcome not driving a car in a hypermobile society through documenting and classifying their solutions. Resilience is a term often used to describe an individual’s capacity for overcoming an adverse or difficult situation [26]. In gerontology research, it tends to focus on the individual and how far resilience can be built within an individual or whether it is more or less part of someone’s personality [27]. Indeed, in previous research, some older people successful in giving up driving talked about their openness and willingness to change (from driving) as being a major part of the success [4]. This fits current Western political ideology of reducing top-down governance, and the growing importance of individuals being able to look after themselves. Although this is likely to be part of the solution, other research suggests that taking an individualistic view of resilience negates the role and support of others within this. For example, the informal support network of family and friends [25] and the role of the state in sharing out a finite and competitive resource, such as the transport network [18] are both important in supporting people who have given up driving. To this end, this paper uses Bourdieu’s theory of capital as a starting point, as a means of categorising different ways older people cope without a car.

Bourdieu argues that people have different levels of three forms of capital: social, cultural and economic capital, that are seen as a resource used in maintaining health and wellbeing [28]. Social capital is defined as access to contacts, friends and family and other informal support networks through mutual sharing of resources, skills, support and reassurance. Research suggests social capital is related to many aspects of civil society, including health, crime, economic growth, accessing job markets and education [29,30]. Researchers have typically argued whether social cohesion or social networks are different to social capital, as the terms are often used interchangeably. In this paper, we will identify any differences within the category if possible. Cultural capital is often described in three ways, as educational attainment, possession of goods and people’s values, skills, knowledge and tastes [31]. Economic capital is described as material assets including, for example, financial resources, land or property ownership [31]. People have varying levels of capital, and they are different across different places, with people from different backgrounds, and can vary across the lifecourse, which can cause inequalities. There is limited research on capitals and mobility for people in later life. However, Gray et al. (2006) noted that strong local social capital is important for mobility in rural areas for those without access to a car [32]. Shergold and Parkhurst also noted the importance of social capital in supporting mobility in rural areas across a range of journeys [33]. Shergold et al. [34] applied the concept of motility [35,36] to rural-dwelling older people, motility brings together social and mobility resource making mobility easier, including knowledge and understanding of the transport network, norms (cultural capital), availability and support from others (social capital) and ability to pay (economic capital). The notion of capital is a somewhat contested term, exact definitions have never been agreed upon, and measurements remain troublesome [32], but in this paper it is used to categorise barriers and enablers to mobility for older people, as a basis for development of a theoretical model of mobility capital.

## 2. Materials and Methods

### 2.1. Design

Semistructured interviews were conducted with 36 participants aged over 65 years to explore their travel and transport needs. The participants were placed in four different groups: (1) older people who still drive; (2) public bus and community transport users; (3) nondrivers who regularly rely on friends and family (outside the immediate household) to give them lifts and; (4) people who walk. The interviews were complemented by four focus groups with individuals within each of their groups. All but one individual from a walking group who was unwell took part in the focus groups. Focus groups were chosen to help people to think more widely about their daily mobility and transport use, comparing and contrasting issues and solutions, and also to help generate new ideas and discuss new technologies around driving, which might otherwise be hard individually.

### 2.2. Participants

Participants were sought through the research network of older people in South Wales, United Kingdom, answering an advert for people who fit one of the four categories. The research network holds a record of 600 older people and organisations and groups of people who have indicated that they are interested in taking part in research, possibly through taking part in previous research or attending a local research seminar or discussion hosted by the network. The advert was sent out by email and was subsequently displayed at local events run by the network. It invited anyone over the age of 65 living in South Wales to take part, not mentioning health or wellbeing. The advert noted that a discussion would take place on everyday transport mobility, including motivations for transport, examining the importance of mobility as well as the barriers and enablers. People were selected to take part on a first come, first served basis, filling each category until there were nine in each (a good number for focus groups). People were placed into one of the four respective transport categories if they used that mode most often for their journeys, they did not need to be exclusive users of that mode. A decision was made, using an initial screening question when approaching the individual by email or phone to arrange the interview. Only one person changed group to (4) walking, from (3) rely on lifts from family and friends after the interview determined he was more of a walker. This resulted in an additional “walker” and one less “reliant on lifts from family and friends”.

A total of 36 participants took part, with a range of ages from 65 to 92 and an average age of 73.2 years, 25 were cohabiting with a partner and 11 living alone (see Table 1). Overall they were mostly healthy individuals, although this was not a selection criteria but rather a result of who came forward. Participants were asked to self-report their health on a scale from 1, very poor to 9, very good; an average of 7 on the scale was found overall. The highest average, indicating “best” average health, was among the people who walked, and the lowest average was among the people getting lifts from family and friends.

### 2.3. Procedure and Tools

Individuals were asked where they would like interviews to take place, in a location where they would feel most comfortable. A total of 14 interviews took place in participants’ homes, following lone working protocol, with 12 taking place in public spaces, including five in coffee shops, four in a local community centre and two in a local public house. The discussion was kept fairly open, and participants were free to talk around two key themes: (1) current mobility and meanings of that mobility to the individual and their life and; (2) barriers of that mobility, examining how they overcame such barriers. Focus groups lasted around an hour long and took place at a local community centre around 2–3 weeks after the interviews, these picked up key points from themes from the interviews but also shared solutions to overcoming barriers and worked towards co-producing solutions to overcoming the barriers and maximising enablers. In both the interviews and the focus groups, two techniques were used:

Apprenticing [37] was used to examine current transport motivations, needs and behaviour, along with explaining the barriers and enablers to the transport that they use. This technique gets participants to talk the researcher through a particular journey that they make regularly. It is done in a high level of detail, with the idea that the researcher could repeat the journey for themselves and identify the barriers and enablers themselves.

Abstraction [37] techniques were used when discussing overcoming issues to everyday mobility if their experience was different. It involves three types of questions: (a) counterfactual, (b) othering and (c) future scenario testing. (a) Counterfactual detail asks the individual what would happen if things were different for them, for example, if their health or individual circumstances changed. (b) Othering is a technique used to get participants to discuss whether their experiences (and those of other people in the focus groups) would be the same for other people or whether they are unique to the individual. (c) Future scenario testing examines potential changes to the future and then discusses how they might alter their ability to get out and about. These included autonomous vehicles, mobility as a service, use of technology in active travel and public transport, and enhanced taxi services on demand.

### 2.4. Analysis

The interviews and focus groups were digitally recorded, transcribed verbatim and thematically analysed. The interviews were analysed first to inform the structure of the focus groups discussions around barriers and enablers. For both thematic analyses, the researcher identified key themes in each transcript, created a long list of codes, then after all interviews and then all focus groups had been analysed, reviewed all codes together, collapsing codes and generating overarching themes and subthemes. Etic (stemming from themes derived from current theory, models and literature) and emic (stemming from analysis of the data itself) coding was employed during the analysis. Etic codes looked to place the data within the three capitals, social, cultural and economic. Emic codes came from the interpretation of the data itself. That is, data that might modify or challenge this model were analysed and additional, previously undiscovered codes and themes were added.

## 3. Results

From the initial model of social, cultural and economic capital, the first two categories were distinctly noticeable in the findings. Two additional categories were added, one of which encompasses economic capital in this instance, but goes wider than just being monetary based. The provision of infrastructure and services themselves was frequently mentioned, and it was deemed this could become an additional category that also contained economic capital. Termed infrastructure capital, this was having the necessary resource to carry out the mobility that is provided in the local areas and is somewhat out of the individual’s control. Finally, individual resilience was also described, and this is included in an additional category, as again it does not quite fit the categories of social, cultural, economic or infrastructure.

### 3.1. Infrastructure Capital

The category with the most response within, both in terms of depth and description but also most frequently mentioned belonged to infrastructure. Older people mentioned how important it was that there had to be the right provision of services, support and hard infrastructure for individuals to be able to access transport beyond the car. Economic capital forms part of this, especially with regard to public bus use in the UK, as it includes free use of buses,

“Well luckily the buses are free. Makes a huge difference. I wouldn’t go out half as much as I do.” (female, aged 77, bus user)

Older people who had good-quality bus services were far more likely to be able to give up driving with fewer problems, but there was a noticeable feeling of being lucky to be on a bus route,

“Well we’re lucky, lucky for now at least, we’ve plenty of buses get us into town and that’s where we want to go and the other way they go on to the coast, so we can go there too.” (female, aged 77, bus user)

And that some people were “out of luck” by not being on a bus route, especially if it had had buses taken off that route,

“They reduced the service down and down and then nothing. One day just took it away. We’re on a Sunday only route but I want to use the bus everyday.” (male, aged 81, lifts from friends and family)

Poor services running on routes were noted, especially around the time of the day services ran, sometimes people wanted to travel at very different times and frequencies than the service provided,

“We are on a bus route, but we can’t use it, well not often, in fact we don’t tend to at all, because there is just one service in and one service out of the city. The one service in is like 8am and one service back isn’t until 4pm. We might, well usually we just want to be there for an hour or two you know.” (female, aged 83, lifts from family and friends)

Car users predicted that the provision of services could never match up to their ability to drive when and where they wanted and often this worried them,

“I go loads of places in my car. It’s so convenient. I mean what am I going to do if I have to give-up. The buses won’t go anywhere near where I want them to go! I’ll be stuck.” (male, aged 80, car driver)

The service was also seen as poor if there were concerns about the bus driver, the lowering of the floor of the bus to accommodate easy access and not being enough room on the buses,

“It’s the quality of the buses really. The actual bus service too. Whether it stops low enough to get on without a step. And the driver doesn’t drive off before I sit down. Big barriers to me!” (female, aged 77, bus user)

Similarly, with regard to walking, the importance of having quality infrastructure to be able to walk on is important, including pavement being in good condition,

“There is a pavement, but it is in such bad state of repair, I have to walk on the road but I hate that.” (female, aged 80, walker)

Crossing the road was a frequently mentioned issue for older people,

“There is a busy road I always struggle with, it’s walking round a long way to the crossing or taking your life in your hands. At my age you can’t do that.” (male, aged 77, walker)

Not having enough room to walk was often mentioned too, for example, sharing space with other users or items,

“The pavement is used by so many other things, cyclists, bins, parked cars, it’s just too narrow.” (female, aged 81, walker)

### 3.2. Social Capital

The next most frequently and vociferously mentioned answers belonged around social capital. Older people relied very much on the support of others to help them with literal mobility, replacing journeys they would have made driving themselves with journeys as passengers in other people’s cars. This was easy when someone was travelling to the same place anyway, and this had a sense of altruism, of helping others associated with it, which was enjoyed by the driver,

“I pick up two people to take them for church on Sunday morning. I don’t mind. I like to provide them with help. Goodness knows they wouldn’t be able to get there without it. And it’s important to them.” (female, aged 73, car driver)

This was not necessarily always from someone well known to the individual, especially if traveling to the same location. Sometimes a neighbour or someone in the community offered this support. It was noted how this created a sense of friendship for some,

“I’m so grateful for [‘name’] for giving me a lift, we’ve become close friends. We realised we lost our husband around the same time from a similar illness. So yes a source of friendship and yes comfort.” (female, aged 82, lifts from family and friends)

However, on other occasions, having family or close friends nearby resulted in providing practical support, which was very welcome,

“Thank goodness I have my daughter living nearby, otherwise I’d go stir crazy not going out.” (female, aged 86, lifts from family and friends)

However, receivers of literal mobility often felt very guilty of being a burden on others and used techniques to deal with that, including reciprocation,

“I pay back my lifts. When my son or daughter-in-law give me lifts, I’ll make sure I’ve baked them a cake or paid my way somehow.” (female, aged 86, lifts from family and friends)

Although usually the support given for mobility was welcomed, people still lamented that their mobility was not as good as it once was when they were drivers,

“Yes I get lifts now all the time, but it isn’t the same, it’s hard to explain, I suppose you can’t go when you want. I don’t go when I want now.” (female, aged 80, lifts from family and friends)

Support was also linked to nonliteral mobility, helping others to get things in that they would have got using their car, but that they can’t now they walk or use the bus. This was often related to a sense of good feeling within a community or neighbourhood,

“What’s lovely ‘round here is the help I get. I mean my neighbour, he makes sure I’m ok. He gets me in a pint of milk, or something I can’t or struggle to carry on the bus.” (male, aged 80, bus user)

This was also noted in that some areas seem to have more support than others, though discussions on this were more anecdotal than discussing people’s own experiences,

“It wouldn’t have happened back where I lived, it wouldn’t. People didn’t know each other, so they didn’t look out for each other at all, didn’t care, here its different.” (female, aged 77, walker)

It was, though, rather rare for the participants to talk about receiving emotional support from others in giving up driving,

“I think I went 6 months through some kind of grieving process when the doc took away my keys. I didn’t seek help but I also didn’t get any sympathy. You’re expected just to get on aren’t you, now you’re old.” (male, aged 80, walker)

It was also rare to find lifts given out for what might be termed discretionary activities, for mobility for its own sake,

“I can’t get out and about so much like I used to. I miss going to the sea for an ice cream by the beach. There is no bus there. I can’t really ask my busy family!” (female, aged 80, bus user)

Much of the social capital stems from “soft” support, within family and friends and within communities, though it was noted by a few participants that social support within informal or even formal groups can help with mobility,

“Having a walking group got me walking again. I’ve always been interested, always enjoyed walking, but I’d got out of it and I thought I was too old and fat, but they, well we, support one another and now I walk everywhere.” (female, aged 76, walker)

### 3.3. Cultural Capital

The final two categories both received around equal attention from older people and supported the social and infrastructure capital. Cultural capital stems from the surrounding environment, whereas individual capital comes from within.

Sometimes cultural capital created a frustration, rather than a resource to draw upon. The car was seen as a norm in society, and people missed doing the journeys they used to do and had got used to doing and were seen as part of the normal everyday, and this had negative affective consequences for the individual,

“to think I can’t even get to the hospital easily just makes me feel a failure.” (male, aged 80, bus user)

The opposite was true for people who had been less wedded to their car throughout their lifecourse, for example, people who had been highly multimodal,

“Well I’ve been using the bus to get to town now anyway for a while. So I’m used to it.” (female, bus user, aged 81)

Also, there was a feeling among the participants that different locations could have different cultural capital,

“well if I lived in London it would be different. I lived there in the 90s. There is so much public transport. And you know everyone uses it. No one really has a car you know.” (male, aged 86, bus user)

A hierarchy of norms of use appeared among the participants, car use was seen as the most normal everyday form of transport, with driving at the top followed by being a passenger, followed by using a bus, but then walking and cycling were seen as going against the norms of society for older people,

“Yeah walking is still not normal especially for doing things. I mean part of it is the practicalities of carrying stuff but also people don’t walk now do they. Not unless it’s for leisure.” (male, aged 72, walker)

Another gap in cultural capital noted by participants was the lack of being involved in decisions affecting their transport systems,

“Look I don’t know if it is the kind of thing that would do any good, but I, we, don’t have any say in the transport we have around here. We need to know when services are stopping and we need to stop them doing it. It’s odd they ask us our opinion on loads of stuff but not transport. Like a secret brotherhood.” (male, aged 82, bus user)

### 3.4. Individual Capital

There were incidences where the individual’s own experience, attitude or values supported mobility without a car. Being able to use a mode of transport requires some ability to use it, knowledge of how to use it and understanding of the norms, and older people discussed this with regards to buses,

“Well I’ve always used the bus to go into town, it’s just easier, then when I had to use more buses because we’d given up driving, it wasn’t so much of a shock really.” (female, aged 88, bus).

With regards to walking, there is a similar need to be able to walk but also to be able to be personally resilient in obstacles, such as a feeling of needing good balance or good hearing and observation skills,

“To get over all the cracks in the pavement you need great balance, I haven’t always got that now, I’m worried about falling.”(female, aged 80, walker)

“You need good hearing and need your wits about you. Cyclist come up right behind.” (male, aged 70, walker)

Some older people talked about having a personality that meant they were open to change, which helped when giving up driving, openness to trying new forms of transport,

“I saw it as a challenge, and it was a challenge I can tell you. But I challenged myself to using the bus. I hadn’t used them for years, so I was intrigued, and I’ve actually enjoyed it. Haven’t looked back.” (male, aged 80, bus user)

Similarly, some older people talked about being open to visiting new places,

“Well you can’t actually go on the bus to where I’d normally do my shopping, so I now go somewhere different, somewhere I wouldn’t normally have gone and I actually have enjoyed the change.” (female, aged 75, bus user)

## 4. Discussion

In terms of importance, infrastructure capital, drawing on provision of services, hard infrastructure and economic costs of transport provision seem to be the highest priority for older people. It was this capital that contained so many barriers and enablers. Where this capital was found in abundance, older people felt able to cope with the loss of giving up driving. The provision of bus services, of good walkable neighbourhoods placated the negative side effects of not driving. These included having good-quality bus services going to a range of places, or at least where people want them to go, on good-quality, easily accessible buses, with driver sympathetic to the needs of older people, as has been suggested as important in previous research [22,23]. The provision of quality pavements including well-maintained, continuous surfaces segregated from other users, with good crossing facilities helped people stay mobile, again similar to previous research [8]. However, these alone are not enough. It is unlikely the same levels of mobility or of need can be accessed through infrastructure alone, and it is likely social capital is needed to plug many gaps in provision. Drawing on friends and family was often cited as plugging mobility gaps by provision of literal mobility, as noted in previous research [4,25]. Again, there are noticeable gaps in social capital, especially emotive and discretionary activities not being fulfilled, as have been noted previously by Murray and Musselwhite [25]. Also, drawing on both these levels of resource can be supported by cultural capital by having norms surrounding use of public transport and of walking, often which were missing, according to the older people in this research. Previous research [5,6,8] noted that the car dominates norms in society, with older people particular excluded, who give up driving. There was also a feeling that outside of major cities, especially outside of London in the UK, having cultural capital in the form of support from age-friendly policies and transport policies in general were lacking. Finally, these levels of resource also require an amount of individual capital, an ability to be able to use public transport, to know how to use it and to understand how to operate it, similarly, an ability in being able to walk and a desire to take part in something new and potential for changing travel behaviour. Having a gap in any of the four levels of capital reduces the ability for older people’s mobility when they give up driving. Figure 1 outlines the relationship between the capitals, showing how infrastructure is of highest importance to older people, followed by social capital, supported by cultural and individual capital.

Capitals are not always a resource to draw upon in a positive sense. The norms of a society geared around the car, in line with previous research [8], mean not using a car in later life leaves a big gap in cultural resource. It is a particular concern to older people in that mobility through driving a car represents a feeling of youthfulness and belonging and being part of society [5,6]. In terms of social capital, it cannot always be drawn upon in full due to feelings of being a burden and difficulties or challenges in reciprocation, which has been documented in previous research [4,25]. In addition, the social capital does not always deliver exactly the levels of mobility expected and wanted. In particular, emotional support of social capital is not always evident, despite being in demand, as has been found similarly in previous research, practical support is easier to provide than emotional support [25]. Also, how social capital does not plug the gap of journeys for discretionary purposes or for their own sake, as previous research has found, people felt they can ask for a journey to a hospital but not to a coffee shop or to the seaside, and older people really miss such journeys [38,39].

In terms of the gap in cultural capital where transport policies and practice do not cover ageing populations well, there is a need perhaps for more involvement in social capital. Older people generally felt they would like more involvement in the shaping of mobility services in their local areas. There is a chance to utilise social capital and move it in the direction of cultural capital, to create transport action groups among older people to feed into transport planning. Public involvement in services, through co-development for example, is beginning to take shape in redesigning urban areas, for example, collaborative, redistributive urbanism and intergenerational urbanism [40,41], and transport needs to follow suit.

Capitals not only vary between people, but vary from place to place. This is very much in evidence for cultural capital, where norms can help people be more mobile in some places than in others. In cities, use of public transport, especially buses, is higher than in rural areas, some of this is provision, but it also effects norms through ownership of cars. Those who have got used to using alternative transport to the car have higher levels of individual capital in skills and knowledge, but also this is greater if cultural capital in terms of norms supports these skills and abilities. This is similar to being “public transport” ready, as noted through motility [34]. Women tend to have higher levels of public transport knowledge, having been more multimodal throughout life than men, who are more wedded to their car [8]. This helps them in later life when giving up driving. There did not seem to be any social capital differences between men and women and support groups. Both groups were equally able to use social capital to help when driving had ceased, both drawing on family and friends if they could, but also on local community support. People attributed the ability to draw on wider community support from people outside of their family and friend networks on having a cohesive and friendly local area to live in. This links to a social cohesive element on social capital. It is beyond the scope of this paper to be able to identify if some areas have higher social cohesion than others, but it is certainly noted as a thread stemming from the participants themselves.

Although this was a project carried out with relatively few people in one geographic location in the UK, making it hard to generalise to different contexts, the in-depth nature of the research adds extra dimensions to the findings, noting variance between and within individuals and their capitals. It would be useful to now ascertain if this model is present in other contexts, especially in places where it might be assumed capital might vary hugely, for example different geographical locations and in different cultures. It might also be useful to enumerate some of the findings to address how to overcome gaps in capital, helping to prioritise the best interventions. How far the capitals might change in future ageing populations will also be interesting. How will an increase in older people who are more wedded to their cars than ever cope with giving up driving in later life, as will be found in western societies over the next 20–30 years? How will changes in transport provision, mobility as a service, potential of increased sharing mobility services or the promise of increasingly automated vehicles alter the capitals, for example? Finally, changes in the nature of work for older people, potentially working later on in life, will alter the need for mobility and transport, and in turn affect how people will draw on capitals.

## 5. Conclusions

This paper offers a new conceptual framework built around capitals to try and identify how different levels of capital resource can support people when they give up driving. It identifies four different types of capital that are interconnected. Most importantly to older people is infrastructure (provision of services and of built environment structures to support public and community transport and walking and cycling), followed by social capital (people, friends, family, social cohesiveness of community and neighbourhoods), which is supported in turn by cultural capital (norms, expectations and rules) and individual capital (skills, abilities and openness to change). The findings suggest that interventions aimed at improving age-friendly transport should start with infrastructure capital, to make spaces for walking, for cycling, to provide quality alternatives to the car through public and community transport. This is followed by allowing social capital to flourish, to pool resources to encourage community and neighbourhood solutions to transport. One way of bridging the gap to improving cultural capital is to develop and support bottom-up community-based transport groups, to get people involved in shaping their own transport solutions, something very much wanted by older people themselves. Finally, supporting individual’s to be more transport-ready, improving their motility, encouraging use of public transport and maintaining walking and cycling throughout the lifecourse so they have the skills and abilities to use such provision through and into later life is needed.

## Figures and Tables

**Figure 1 ijerph-16-03327-f001:**
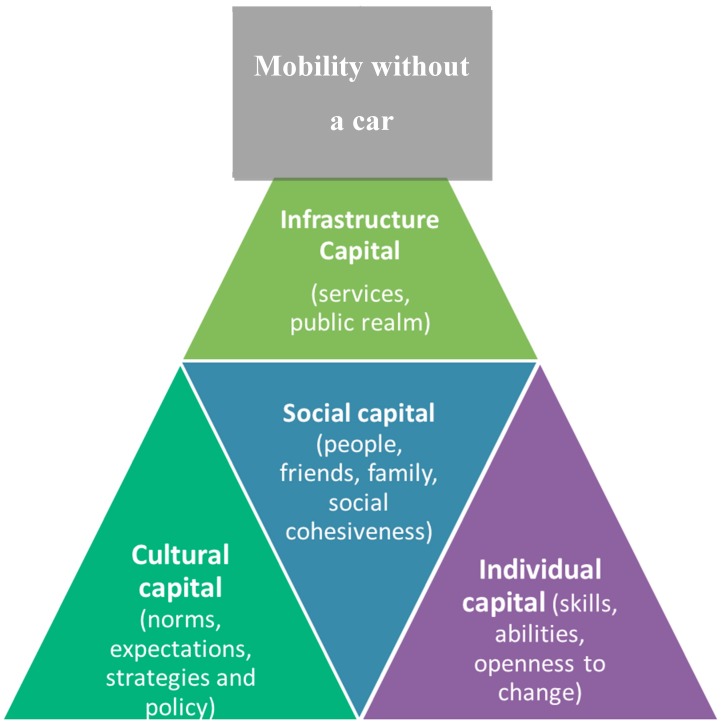
Theoretical model of mobility capital, showing the relationship of different capitals to enable mobility in later life for those not driving.

**Table 1 ijerph-16-03327-t001:** Background of participants in the study by type of transport commonly used.

Participant Group Based on Most Common Mode of Transport	*n*	Age Range (Average)	Living Arrangement	Health (Self-Score 1 = Poor to 9 = Good)
Focus group 1: Drivers	9	65–87 (73.9)	In couple, = 8 On own = 1	6.7
Focus group 2: Bus users	9	65–86 (71.7)	In couple = 7 On own = 2	6.5
Focus group 3: Lifts from family and friends	8	72–92 (78)	In couple = 4 On own = 4	6
Focus group 4: Walkers	10	65–81 (70.2)	In couple = 6 On own = 4	8.2
Total	36	65–92 (73.2)	In couple = 25 On own = 11	7

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
