# Peer review of "Developing A Model of Mobility Capital for An Ageing Population"

_ijerph, 2019, doi:10.3390/ijerph16183327_

Round 1

Reviewer 1 Report

An interesting and relevant paper examining mobility among the elder population.

Generally the paper is well written and includes some interesting insights from empirical data. It requires attention in a number of areas before it is ready for publication. The paper would benefit from significant editing as some sentences are incomplete or have grammatical/spelling errors or are simply not clear e.g. lines 82/3, 91, 97, 1-2, 105, 113, 224/5, 252,  273,

The introduction would be more useful to the reader if it were to include a general and punchy introduction before being followed by the literature review. The review of Bourdieu’s capitals is too brief and does not fully set the reader up for the empirical data that follows. This lack of detailed overview follows in the presentation of the results which are presented in a nearly raw form with little interpretation and engagement back to the key themes of the literature.

The analysis section is written in a rather clumsy manner and would benefit from a re-write. I would like to see more development of the discussion as it is very general and is not sufficiently developed.

The authors’ conclude: ‘One way of bridging the gap to improving cultural capital is to develop and support bottom-up community-based transport groups, to get people involved in shaping their own transport solutions, something very much wanted by older people themselves’. This is something that has been well known for some time, and thus begs the question – what is the contribution to knowledge that this paper is making? I have no doubt that the authors’ will assert that it is 4 forms of capital, but given that this is underdeveloped it remains unconvincing as a new conceptual model.

Author Response

An interesting and relevant paper examining mobility among the elder population.

Thank you.

Generally the paper is well written and includes some interesting insights from empirical data. It requires attention in a number of areas before it is ready for publication. The paper would benefit from significant editing as some sentences are incomplete or have grammatical/spelling errors or are simply not clear e.g. lines 82/3, 91, 97, 1-2, 105, 113, 224/5, 252,  273,

Significant editing carried out throughout the work to address incomplete sentences, grammatical and spelling errors. The sentences in the lines noted are given attention and changes made.

The introduction would be more useful to the reader if it were to include a general and punchy introduction before being followed by the literature review.

Introduction separated into 1.1. Context and 1.2. Literature

The review of Bourdieu’s capitals is too brief and does not fully set the reader up for the empirical data that follows.

The review of capitals is only meant as an outlining framework. Slightly more detail is included for the reader. References are included for more detail.

This lack of detailed overview follows in the presentation of the results which are presented in a nearly raw form with little interpretation and engagement back to the key themes of the literature.

This is deliberately done to show the readers how the data fit under the categories. Much analysis went into sorting the data into the categories shown.

The analysis section is written in a rather clumsy manner and would benefit from a re-write.

Analysis section has had a re-write.

I would like to see more development of the discussion as it is very general and is not sufficiently developed.

Overall, we have related our findings to previous research and drawn on links between them as appropriate.

The authors’ conclude: ‘One way of bridging the gap to improving cultural capital is to develop and support bottom-up community-based transport groups, to get people involved in shaping their own transport solutions, something very much wanted by older people themselves’. This is something that has been well known for some time, and thus begs the question – what is the contribution to knowledge that this paper is making?

We are unaware of much literature that support this is the case in transport? No one has placed barriers and enablers of mobility around a theory of capitals previously, this is new.

I have no doubt that the authors’ will assert that it is 4 forms of capital, but given that this is underdeveloped it remains unconvincing as a new conceptual model.

It is the basis of a theoretical model, the next stages will be to test and modify or enumerate the model to see how it works. This is noted in the conclusion section.

Reviewer 2 Report

Comments to the author

The authors described the results of their study titled; Mobility capital for an ageing population

The aim of their study was to assess older people’s relationship with everyday mobility;. examining the complex and changing nature of mobility, wellbeing and quality of life, through interviews and focus groups with older car drivers, community and public transport users and those that walk for the majority of their journeys.

This subject is interesting. However the way the study is performed and conclusions are made raise large concerns. Next the paper has a low quality and lacks structure.

Title; In the title the kind of study is lacking. What kind of study do the authors describe? Is it a cross sectional study? Is it a review?

Abstract; The abstract is not easy to read and lacks a clear structure. It is not very clear what is done and what can be concluded. This should be more clearified .

Introduction; The introduction is too long for this subject. It can be reduced and more structured. For example; Line 30-37. As in the meantime everybody knows that the worlds population is aging this can be much shorter.

Materials';

36 participants is a very small group for this subject. I would expect far more participants.

Participants were sought through the research network of older people in South Wales, United 140 Kingdom, answering an advert for people who fit one of the four categories. What kind of peaople are that? Is it a certain group? This is very unclear.

Next how many responded on the advertism? How many were selected? How were they selected? It is hard to understand what was done.

What I miss is information on how frail these elderly were. Were these al vital elderly? Were they Frail? How frail?

Next,  what was questioned precisely in the structured interview?

Results

In the result section all the answers of the participants are placed in the text which makes it hard to read as there are many remarks. Maybe this should be placed in a table.

Discussion;

What are the limitations? What are the strengths of the study? What is the clinical relevance? What should readers do with the findings?

Author Response

The authors described the results of their study titled; Mobility capital for an ageing population

The aim of their study was to assess older people’s relationship with everyday mobility; examining the complex and changing nature of mobility, wellbeing and quality of life, through interviews and focus groups with older car drivers, community and public transport users and those that walk for the majority of their journeys.

This subject is interesting. However the way the study is performed and conclusions are made raise large concerns. Next the paper has a low quality and lacks structure.

Title; In the title the kind of study is lacking. What kind of study do the authors describe? Is it a cross sectional study? Is it a review?

We wouldn’t normally include the type of study in the title, as a concession I rename the title what the paper attempts to do, “Developing a model of mobility capital for an ageing population”

Abstract; The abstract is not easy to read and lacks a clear structure. It is not very clear what is done and what can be concluded. This should be more clarified .

Abstract has been re-written

Introduction; The introduction is too long for this subject. It can be reduced and more structured. For example; Line 30-37. As in the meantime everybody knows that the worlds population is aging this can be much shorter.

I disagree that this can be shorter. It is imperative to the readership of this article, who include transport and planning experts, that they understand the wider ageing context. They do not necessarily know the growing importance of an ageing population.

Materials;

36 participants is a very small group for this subject. I would expect far more participants.

This is actually very large for a qualitative study and many qualitative researchers may well criticise us for having too many participants. The number of participants in relation to the generalizability of the findings is mentioned as a limitation in the discussion. Please see the final paragraph which opens with,

“Although this was a project carried out with relatively few people, making it hard to generalise to different contexts, the in-depth nature of the research adds extra dimensions to the findings, noting variance between and within individuals and their capitals. It would be useful to now ascertain if this model is present in other contexts, especially in places where it might be assumed capital might vary hugely, for example different geographical locations and in different cultures. It might also be useful to enumerate some of the findings to address how to overcome gaps in capital, helping to prioritise the best interventions”

Participants were sought through the research network of older people in South Wales, United 140 Kingdom, answering an advert for people who fit one of the four categories. What kind of people are that? Is it a certain group? This is very unclear.

Added in, “The research network holds a record of 600 older people and organisations and groups of people who have indicated they are interested in taking part in research, possibly through taking part in previous research or attending a local research seminar or discussion hosted by the network), The advert was sent out over email and appeared at local events run by the network. It asked for anyone over the age of 65 living in South Wales to take part, not mentioning health or wellbeing”

Next how many responded on the advertisement? How many were selected? How were they selected? It is hard to understand what was done.

Added in, “People were selected to take part on a first come, first served basis, filling each category until there were 9 in each (a good number for focus groups). People were placed into each transport category if they used that mode most often for their journeys, they did not need to be exclusive users of that mode. A decision was made using an initial screening question when approaching the individual by email or phone to arrange the interview. Only one person changed group to (4) walking, from (3) rely on lifts from family and friends after interview determined he was more of a walker. This resulted in an additional walker and one less reliant on lifts from family and friends”

What I miss is information on how frail these elderly were. Were these al vital elderly? Were they Frail? How frail?

These were all healthy older people. Added in, “Overall they were mostly healthy individuals, although this was not a selection criteria but rather a result of who came forward. Participants were asked to self-report their health on a scale from 1 very poor to 9 very good; an average of 7 on the scale was found overall. The highest average, indicating ‘best’ average health, was among the people who walked and the lowest average was among the people getting lifts from family and friends”

Next,  what was questioned precisely in the structured interview?

The interviews were semi-structured. Section 2.3 notes the key discussion points which were used as prompts to develop discussion and generate data.

Results

In the result section all the answers of the participants are placed in the text which makes it hard to read as there are many remarks. Maybe this should be placed in a table.

This is standard practice in a qualitative piece of research.

Discussion

What are the limitations? What are the strengths of the study? What is the clinical relevance? What should readers do with the findings?

Limitations and strengths noted in the final paragraph of the discussion.

This is not a clinical paper, so there is no clinical relevance.

The conclusions state what should happen. Added to this are actors who might make this happen.

Reviewer 3 Report

The four capitals framework used in this paper provides as practical, easily understood and comprehensive format for analysis.

The data presented comes from older people currently experiencing challenges with transport and how it influences their daily life and well being (although the numbers of participants are limited and they come from only one specific area of the UK).

The current dominance of the car is illustrated along with the consequences of losing access to private transport. Gender and urban/rural differences are noted. These aspects along with the difference between "necessary" and "discretionary" travel mirror the findings of my own research in New Zealand - see 

Davey, J. (2007) Older people and transport: coping without a car. Ageing and Society, UK.27(1) p.49-65

Author Response

The four capitals framework used in this paper provides as practical, easily understood and comprehensive format for analysis.

The data presented comes from older people currently experiencing challenges with transport and how it influences their daily life and well being (although the numbers of participants are limited and they come from only one specific area of the UK).

Yes, added in final paragraph of the discussion, “in one geographic location in the UK,”

The current dominance of the car is illustrated along with the consequences of losing access to private transport. Gender and urban/rural differences are noted. These aspects along with the difference between "necessary" and "discretionary" travel mirror the findings of my own research in New Zealand - see 

Davey, J. (2007) Older people and transport: coping without a car. Ageing and Society, UK.27(1) p.49-65

Thank you for your very positive review. I have read Davey (2007) previously and you are right, I will certainly include it in our article.

Round 2

Reviewer 1 Report

It is good to see that some of my comments have been taken on board. The paper is somewhat improved. I would stand by some of my original comments which the authors have refuted - so I guess we have a different perspective. In any case I believe the paper is acceptable as it now stands.

Reviewer 2 Report

comments related to the abstract; In the abstract is is not clear how this study has been performed and how many elderly were involved. How were these elderly recruited. Where does the study took place? 

Comments related to the introduction;

-  The introduction is very long. Too long I would say. Reading al these information, the reader get lost in the large amount of information. Please try to give information about the problem and then what is known in literature and then what is unknown and should be the subject for the study. Next especially the start of the introduction can be much shorter. We all know by now that there are more and more elderly. One does not need to explain that in depth with numbers.

the aim of study should be at the end of the introduction 

Comments to material and methods; 36 participants is very low. Which questions were asked? Can you provide a table withe the detailed information on the interviews.

Comments to the discussion; please explain why there were only 36 participant.

What are the strengths and the limitations?

Comments to the conclusion; can you make the conclusion short and powerfull?